# Back to Basics: Motion Representation Matters for Human Motion Generation Using Diffusion Model

## Abstract

Diffusion models have emerged as a widely utilized and successful methodology in human motion synthesis. Task-oriented diffusion models have significantly advanced action-to-motion, text-to-motion, and audio-to-motion applications. In this paper, we investigate fundamental questions regarding motion representations and loss functions in a controlled study, and we enumerate the impacts of various decisions in the workflow of the generative motion diffusion model. To answer these questions, we conduct empirical studies based on a proxy motion diffusion model (MDM). We apply $v$ loss as the prediction objective on MDM ($v$MDM), where $v$ is the weighted sum of motion data and noise. We aim to enhance the understanding of latent data distributions and provide a foundation for improving the state of conditional motion diffusion models. First, we evaluate the six common motion representations in the literature and compare their performance in terms of quality and diversity metrics. Second, we compare the training time under various configurations to shed light on how to speed up the training process of motion diffusion models. Finally, we also conduct evaluation analysis on a large motion dataset. The results of our experiments indicate clear performance differences across motion representations in diverse datasets. Our results also demonstrate the impacts of distinct configurations on model training and suggest the importance and effectiveness of these decisions on the outcomes of motion diffusion models.

## 1 Introduction

In recent years, Denoised Diffusion Probabilistic Model (DDPM) has been widely applied in human motion synthesis due to its ability to learn latent data distributions. Human motion synthesis can be mainly categorized into human motion prediction (Holden et al., 2017; Pavllo et al., 2018; Aksan et al., 2021; Guo et al., 2022b; Wang et al., 2023; Hou et al., 2023) and human motion generation (Chang et al., 2022; Shi et al., 2024; Tevet et al., 2022a; Zhang et al., 2024a; Chen et al., 2024; Guo et al., 2020; Karunratanakul et al., 2023; Zhang et al., 2024b). These methods have shown great promise in exploring animation styles and qualities through a controlled generative process.

In human motion synthesis, the Motion Diffusion Model (MDM) is capable of denoising noisy samples to produce clean samples along the time steps iteratively based on a parameterized Markov chain trained using variational inference (Ho et al., 2020). Inspired by image generation, human motion sequences within fixed frames can be encoded as feature maps similar to image encoding. A motion sequence can be represented by continuous pose features per frame. A large amount of work uses conditional diffusion models to implement downstream tasks with user control. A conditional diffusion model conditions the diffusion model for specific tasks. At the heart of these steps is the representation of the motion itself. Going back to basics, investigating motion representations is crucial for determining the factors that affect the quality of human motion generation when using a diffusion model.

A variety of motion representations have been proposed in both motion prediction and motion generation to improve the quality of the synthesized motion sequences. Motion representation is commonly some permutation of joint positions or joint rotations, and occasionally with some other

features (Zhu et al., 2023). These representations often impact the construction of the underlying methodology and the quality of the outcomes. However, the use of a particular motion representation is not standardized. This leaves open a fundamental question: What kind of motion representation is more impactful in human motion generation when using a diffusion model? To answer this question, we investigate six motion representations from the literature: Joint positions (JP), Root Positions and 6D Joint Rotations (RP6JR), Root Positions and Quaternion Joint Rotations (RPQJR), Root Positions and Axis-angle Joint Rotations (RPAJR), Root Positions and Euler Joint Rotations (RPEJR), and Root Positions and Matrix Joint Rotations (RPMJR). We compare the human motion generation performances based on the above motion representations using our framework. The results indicate that the position-based motion representation (JP) outperforms in terms of diversity, fidelity, and training efficiency when training in MDM with $v$ loss ($v$MDM). Compared to rotation-based motion representation (RP6JR, RPQJR, RPAJR, RPEJR, and RPMJR), the 3D coordinates of joints are more straightforward to capture and have great potential in various scenarios. However, continuous rotation representation is superior with regard to motion stability.

Our contributions are as follows: We first explore various motion representation training in $v$MDM and MDM, including position-based motion representation (JP) and rotation-based motion representation (RP6JR, RPQJR, RPAJR, RPEJR, and RPMJR) to assess whether they allow the motion diffusion model to learn the latent distribution of human motion more effectively. Second, we empirically study that the integration of the $v$ loss function can trade off motion generation performance and training efficiency. Similar to SMooDi (Zhong et al., 2024), we retarget a large-scale motion dataset (100STYLE) to the SMPL skeleton to test the robustness of our framework. Third, we can generate seamless, natural human motion sequences efficiently using a diffusion model with a concise motion representation, based on two motion datasets (HumanAct12, 100STYLE, and HumanML3D) that cover limited and extensive motion ranges.

## 2 RELATED WORK

Human motion synthesis is still a challenging problem due to the complexity of human motion data. Human motion synthesis consists of human motion prediction and generation. Human motion prediction focuses on forecasting future motion sequences based on past motion sequences, considering the presence/absence of current environments. Human motion generation aims to generate realistic and seamless human motion sequences, enhancing utilities for real-world applications (Zhu et al., 2023). Whether in human motion prediction or generation, the motion representation is crucial in determining the quality of synthesized human motions in data-driven approaches. In the meantime, with the development of biped animation technology and the increasing popularity of generative models, considerable strides have been made in human motion diffusion models in recent years. Therefore, we analyze the various motion representations and human motion generative models from related work.

### 2.1 MOTION REPRESENTATION IN BIPED ANIMATION

Motion sequence is commonly represented by continuous motion features along the frame axis. Motion representations can be classified into position-based representation (Guo et al., 2022b), rotation-based representation (Martinez et al., 2017; Zhang et al., 2024b), and physics-based representation (Peng & Van De Panne, 2017). In this paper, we focus on the first two and do not discuss physics-based motion generation.

Joint positions are the 3D coordinates of joints and are universally employed in human motion synthesis (Guo et al., 2022b; Wei et al., 2023). Using only joint rotations to learn in the deep neural network can not synthesize the root translation. Therefore, rotation-based representations require the combination with movement features. Euler angle rotation representation is more intuitive for users than other commonly used rotation representations (e.g., quaternion, axis angle, rotation matrices). However, Euler angle representation suffers from the discontinuity and singularity (Pavllo et al., 2018). While applying rotation features, the motion representation is commonly concatenated with joint positions, joint rotations, and other features. It aims to reduce raw motion data complexity. Exponential map rotations of all joints and global features are used by Martinez et al. (2017). Exponential map rotations of non-root joints, joint positions, and some velocity-based features are applied by Hou et al. (2023). PFNN predicts future poses represented by joint rotations in expo-

nential map rotation representation and other features, given the user control signal to implement character control (Holden et al., 2017). Axis-angle rotation representation is applied in a series of SMPL models (Pavlakos et al., 2019). Quaternet uses quaternions as the rotation representation because this work supposes the exponential map suffers from problems similar to the Euler angle rotation representation (Pavllo et al., 2018). 6D continuous representation is proposed by Zhou et al., highlighting the limitations of 3D and 4D rotation representations (Zhou et al., 2019). Recently, 6D rotation representation is widely used in motion generation (Tevet et al., 2022a; Chen et al., 2024; Zhang et al., 2024b; Li et al., 2022; Petrovich et al., 2021).

## 2.2 Human Motion Generative Models

Recent human motion generative models can generally be classified into three categories: GAN-based motion generative models, VAE-based motion generative models, and diffusion-based motion generative models. MoDi draws inspiration from StyleGAN (Karras et al., 2020) to synthesize high-quality motion sequences using quaternion rotation representation (Raab et al., 2023). VAEs are also widely adopted in generative models. ACTOR proposes a Transformer VAE to synthesize human motion sequences conditioned by action labels (Petrovich et al., 2021). Since Ho et al. propose denoising diffusion probabilistic models and achieve high-quality image generation (Ho et al., 2020), diffusion models also gain broad traction in human motion synthesis.

The human motion diffusion model is a generative approach that employs diffusion processes to synthesize realistic and natural human motion sequences. Building upon the principles of diffusion models, motion diffusion models iteratively denoise pure noise into human motion sequences, effectively capturing the complex dynamics of human motion data. To validate whether motion representation and loss function design have an impact on the generation results, we focus on these details in the related work. In long-term motion synthesis, long motion sequences can be generated using conditioned diffusion models auto-regressively (Zhang et al., 2024b; Shi et al., 2024; Chen et al., 2024). Joint positions, 6D joint rotations, and foot contact labels are used as motion representation in TEDi (Zhang et al., 2024b). Similar to TEDi, joint positions and 6D joint rotations are also used as motion representation for real-time character control (Shi et al., 2024; Chen et al., 2024). For downstream tasks, motion is represented by the concatenation of joint positions, joint rotations, and velocity-based features (Tevet et al., 2022a; Andreou et al., 2024) according to the work proposed by (Guo et al., 2022a). Music inputs with a text-conditioned motion diffusion model used joint positions as motion representation (Dabral et al., 2023). The synthesis of dancing motions conditioned by music is based on exponential map rotation representation (Alexanderson et al., 2023). In loss function design, most related works use the loss function based on noise prediction (Chang et al., 2022; Zhang et al., 2024a; Shi et al., 2024) or motion data prediction (Tevet et al., 2022a; Chen et al., 2024). Although significant progress has been achieved in the above research, a few studies have focused on the configuration of the motion diffusion model. The quality of generated motion sequences depends mostly on motion priors or complex motion representations.

# 3 Methodology

Our goal is to generate diverse and seamless human motion sequences from pure noise. As shown in Figure 1, our framework is based on MDM for empirical studies on human motion generation using diffusion models. In the forward motion diffusion process, we add noise to the processed motion data, incorporating various motion representations at each diffusion time step. Then, we train a Transformer-based neural network to denoise the noisy motion sequences in the reverse motion diffusion process. During inference, we recover clean motion sequences from Gaussian noise using the trained denoiser and implement temporal smoothing to motions by applying a Gaussian filter.

## 3.1 Motion Representation

Motion representation of a motion sequence $X = \{x_1, x_2, \ldots, x_N\} \in \mathbb{R}^{D \times N}$ is composed of continuous pose features within $N$ frames. Each pose $x_i \in \mathbb{R}^D$ can be represented by a $D$-dimensional feature vector at the $i$th frame, mainly consisting of joint positions, joint rotations, and other auxiliary features. To this end, we calculate the joint positions and five kinds of joint rotations from motion datasets. Joint positions and joint rotations are commonly used in the motion representation

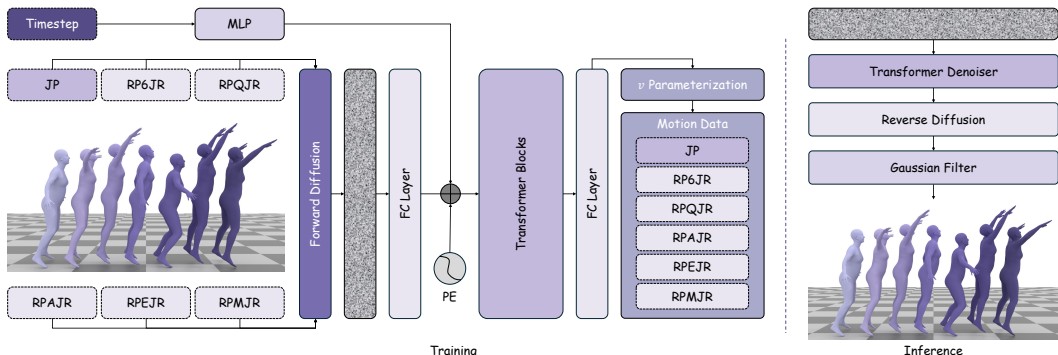

Figure 1: To test the importance of various motion representations, the framework of $v$MDM consists of two stages: training and inference. In the training stage, we first encode the clean motion sequences within $N$ frames to six motion representations (JP, RP6JR, RPQJR, RPAJR, RPEJR, RPMJR). Second, we procedurally add noise to the processed motion data using a forward diffusion module and get noisy motion data after $T$ diffusion time steps. Third, we train a denoiser using a Transformer architecture, and our objective prediction is $v$ parameterization. Based on $v$ prediction, we can recover motion data with a motion representation consistent with the input provided to the denoiser. During the inference stage, our input to the Transformer denoiser is pure Gaussian noise. We then apply the reverse diffusion module and a Gaussian filter to generate motion sequences.

of human motion synthesis. The motion representations we use include two categories: position-based motion representation (JP) and rotation-based motion representation (RP6JR, RPQJR, RPAJR, RPEJR, RPMJR). Additional information can be found in the Appendix A.1.

## 3.2 MOTION DIFFUSION MODEL

As generative models advance, diffusion models have been increasingly applied in human motion synthesis. The motion diffusion model aims to train a neural network to generate diverse human motions from pure noise. In this paper, we focus on the human motion diffusion model to investigate the capacity of learning data distribution based on various motion representations. The motion diffusion model consists of a forward motion diffusion process and a reverse motion diffusion process.

*Forward Motion Diffusion Process.* Firstly, we use different motion representations to encode the motion sequences (i.e. $X = \{x_1, x_2, \ldots, x_N\}$). We then gradually add Gaussian noise using the noise scheduler to the motion data with various motion representations throughout the diffusion time steps $T$. Finally, the noisy motion data can approximate an isotropic Gaussian distribution. We denote the real human motion data distribution as $q(X_0)$. In the forward diffusion process, the Markov noising process for each time step can be formulated by:

$$q(X_t \mid X_{t-1}) = \mathcal{N}(X_t; \sqrt{1 - \beta_t}\, X_{t-1}, \beta_t I) \tag{1}$$

where $\beta_t$ is obtained from a known variance scheduler ($0 < \beta_t < 1$) and $t$ is a single diffusion time step. Given reparameterization of Gaussian distribution, we can sample $X_t$ from $X_{t-1}$:

$$X_t = \sqrt{1 - \beta_t} X_{t-1} + \sqrt{\beta_t} \epsilon_{t-1} \tag{2}$$

where $\epsilon \sim \mathcal{N}(0, I)$. At the end of the noise adding process, we can get noisy motion data with various representations $\{X_1, X_2, \ldots X_T\}$ along the diffusion time steps $T$.

*Reverse Motion Diffusion Process.* We aim to generate natural motion sequences from pure noise. We can model the forward diffusion process by adding noise to motion data for each time step. Because modeling the denoising process is complex and tricky, a neural network is trained $p_\theta(X_{t-1} \mid X_t)$ to learn the denoising probability distribution $q(X_{t-1} \mid X_t)$.

$$p_\theta(X_{t-1} \mid X_t) = \mathcal{N}(X_{t-1}; \mu_\theta(X_t, t), \Sigma_\theta(X_t, t)) \tag{3}$$

Given the addictive property of the Gaussian distribution, we can get a noisy motion representation $X_t$ at any time step from the initial motion representation $X_0$:

$$q(X_t \mid X_0) = \mathcal{N}(X_t; \bar{\alpha}_t X_0, (1 - \bar{\alpha}_t)I) \tag{4}$$

where $\bar{\alpha}_t = \prod_{s=1}^{t}(1 - \beta_t)$.

Only the mean is taken into account because the variance of the target probability distribution is fixed as an assumption (Ho et al., 2020). Given the property of the Markov chain, we can calculate the mean of the probability distribution of $q(X_t \mid X_0)$ by sampling $X_t$ from $q(X_t \mid X_0)$ to replace $X_0$. The mean of the approximated probability distribution is calculated as follows:

$$\mu_\theta(X_t, t) = \frac{1}{\alpha_t}\left(X_t - \frac{(1 - \bar{\alpha}_t)\beta_t}{\sqrt{\bar{\alpha}_t}}\epsilon_\theta(X_t, t)\right) \tag{5}$$

where $\epsilon_\theta(X_t, t)$ is a noise predictor based on a neural network. According to the above, we can train an objective predictor to optimize the loss between the target probability distribution and the approximated distribution.

During training, we add noise to the motion sequences formulated by various motion representations (i.e., forward diffusion). The noisy outputs are passed through a Transformer denoiser to predict the $v$ parameterization. Given the $v$ parameterization, we can predict motion data itself with the motion representation that matches the input. In inference, we only focus on the reverse diffusion process. We input the Gaussian noise into the objective predictor, and then we can sample the generated motion sequences based on the approximated probability distribution. The sampling process can be formulated as:

$$X_{t-1} = \frac{1}{\sqrt{\alpha_t}}\left(X_t - \frac{1 - \alpha_t}{\sqrt{1 - \bar{\alpha}_t}}\epsilon_\theta(X_t, t)\right) + \sigma_t z \tag{6}$$

where $\alpha_t = 1 - \beta_t$, $\sigma_t$ is the standard deviation of the target probability distribution, and $z$ is the input noise. After denoising $T$ diffusion time steps, we can get the final predicted motion data. Given the prediction, we decode them into the 3D coordinates of all joints. We compute the joint positions using the forward kinematics for rotation-based approaches.

### 3.3 NETWORK ARCHITECTURE

We follow the model architecture of MDM and train a neural network in the reverse motion diffusion process to predict the $v$ parameterization. We introduce this objective in Section 3.4. The neural network adopts the Transformer architecture. First, the timestep of the diffusion model is passed through an MLP to project it into the feature dimensions of motion data. Second, a fully connected layer is employed to encode the concatenation of noisy motion and the embedded timestep. Third, the outputs of the fully connected layer are embedded using a sinusoidal positional encoding scheme. Next, the output with positional embeddings is fed through a sequential Transformer block. Finally, the fully connected layer is used again to project the output into the original feature dimensions.

### 3.4 LOSS FUNCTION

In human motion diffusion models, there are two common options for the loss function. One is to predict noise using the neural network, and the other is to predict motion data itself (Tevet et al., 2022a). The diffusion model, which utilizes a loss function to predict clean data, can achieve better performance, according to research by Ramesh et al. (2022). To leverage noise prediction and original clean data, the $v$ loss function is proposed by Salimans & Ho (2022). The loss function based on the $v$ prediction yields better performance in image generation. Therefore, we combine this loss function with geometric loss in MDM. Our loss function in $v$MDM comprises the following four components: $v$ loss, position loss, velocity loss, and foot contact loss.

$v$ *Loss.* The $v$ parameterization is computed from noise and data, and the formula is as follows:

$$v = \sqrt{\bar{\alpha}_t}\epsilon - \sqrt{1 - \bar{\alpha}_t}X_0 \tag{7}$$

We apply the loss weights to this loss function based on $v$ prediction, as shown in the following formula:

$$\mathcal{L}_v = E_{X_0 \sim q(X_0), \epsilon \sim \mathcal{N}(0, I)}\left[\omega(t) \cdot \|v - \hat{v}\|_2^2\right] \tag{8}$$

where $\omega(t) = \frac{SNR(t)}{SNR(t)+1} = \bar{\alpha}_t$.

*Position Loss.* We follow MDM to compute the position loss. This loss function can be formulated as:

$$\mathcal{L}_{pos} = \frac{N}{N-1} \sum_{i=1}^{N} \|fk(X_0^i) - fk(\hat{X}_0^i)\|_2^2 \tag{9}$$

where $fk(\cdot)$ is the forward kinematics function to convert the joint rotations into joint positions. If the motion representation is JP, $fk(\cdot)$ function will not be applied, and $i$ is the frame index of a motion sequence.

*Velocity Loss.* We follow MDM to compute the velocity loss. This loss function is denoted as:

$$\mathcal{L}_{vel} = \frac{1}{N-1} \sum_{i=1}^{N-1} \left\|(X_0^{i+1} - X_0^i) - (\hat{X}_0^{i+1} - \hat{X}_0^i)\right\|_2^2 \tag{10}$$

*Foot Contact Loss.* We also follow MDM to compute the foot contact loss. This loss function is denoted as:

$$\mathcal{L}_{fc} = \frac{1}{N-1} \sum_{i=1}^{N-1} \left\|\left(fk(\hat{X}_0^{i+1}) - fk(\hat{X}_0^i)\right) \cdot f_i\right\|_2^2 \tag{11}$$

where $f_i$ is the binary foot contact labels of four joints, including left ankle, right ankle, left foot, and right foot.

Our final training loss function in $v$MDM is formulated as:

$$\mathcal{L} = \mathcal{L}_{vp} + \mathcal{L}_g = \mathcal{L}_{vp} + \lambda_{pos}\mathcal{L}_{pos} + \lambda_{vel}\mathcal{L}_{vel} + \lambda_{fc}\mathcal{L}_{fc} \tag{12}$$

where $\mathcal{L}_g$ is the weighted sum of position loss, velocity loss, and foot contact loss, and $\lambda_{pos}$, $\lambda_{vel}$, and $\lambda_{fc}$ are the weights of the loss items.

## 4 EXPERIMENTS

In this section, we provide the details of our experimental methodology and any assumptions made. First, we introduce the two motion datasets that we use in our experiments and the data processing methods. Second, we present implementation and training details. Third, we introduce five evaluation metrics to evaluate the generated motion sequences quantitatively. Finally, we briefly introduce three state-of-the-art methods as baseline models for method comparisons.

### 4.1 DATASET

Aiming to train the diffusion model for human motion generation, we use two motion datasets spanning a wide range of actions from minimal to extensive: HumanAct12, 100STYLE, and HumanML3D. Additional information can be found in the Appendix A.2.

### 4.2 IMPLEMENTATION AND TRAINING DETAILS

We integrate six motion representations, including JP, RP6JR, RPQJR, RPAJR, RPEJR, and RPMJR. We fit the processed motion data with the above motion representations to the motion diffusion model separately. All experiments are trained for 500K steps on an NVIDIA A100 GPU. The hyperparameters of the diffusion model are as follows. For the Transformer denoiser: the latent dimension is 512; the number of layers is 8; the size of the intermediate layer in the Feed-Forward Network (FFN) is 1024; the number of attention heads is 4; the activation function is the GELU function. For the diffusion model: the length of the motion sequence is 64; the diffusion time step is 1,000; the noise schedule is cosine one. During training, the batch size of the data loader is 64, and we apply the Adam optimizer with a learning rate of 0.0001. During inference, we also generate samples on an A100 GPU. To fix the jittering issues, we employ the Gaussian filter to do temporal smoothing. The sampling time for motion inference is about 6 seconds.

Table 1: Performance evaluation on $v$MDM and MDM based on HumanAct12 motion dataset under various motion representations. We train all experiments for 500K steps.

| Repr. | Dim. | MDM | | | | | $v$MDM | | | | |
|---|---|---|---|---|---|---|---|---|---|---|---|
| | | FID↓ | KID↓ | Precision↑ | Recall↑ | Diversity↑ | FID↓ | KID↓ | Precision↑ | Recall↑ | Diversity↑ |
| JP | $(J+0) \times 3$ | 38.60 | 0.56 | 0.72 | **0.81** | **17.92** | **27.87** | **0.34** | **0.73** | **0.88** | **18.28** |
| RP6JR | $(J+1) \times 6$ | 36.42 | **0.39** | 0.68 | 0.73 | 17.90 | 49.29 | 0.59 | **0.73** | 0.63 | 16.04 |
| RPQJR | $(J+1) \times 4$ | 59.16 | 0.84 | 0.69 | 0.56 | 16.75 | 68.97 | 0.77 | 0.68 | 0.68 | 14.49 |
| RPAJR | $(J+1) \times 3$ | 51.92 | 0.73 | **0.73** | 0.62 | 16.20 | 52.15 | 0.71 | 0.69 | 0.60 | 16.46 |
| RPEJR | $(J+1) \times 3$ | 72.49 | 0.94 | 0.68 | 0.58 | 16.32 | 49.63 | 0.51 | 0.65 | 0.60 | 15.18 |
| RPMJR | $(J+1) \times 9$ | **34.47** | 0.40 | 0.72 | 0.67 | 15.91 | 75.50 | 0.87 | 0.65 | 0.60 | 16.08 |

## 4.3 EVALUATION METRICS

To evaluate the performance of the generated motion sequences, we follow Tevet et al. (2022b) to apply the five evaluation metrics. Fréchet Inception Distance (FID) score is an indicator to determine the quality of the generated motions. We also apply the Kernel Inception Distance (KID) score to evaluate the quality and diversity of generated motion sequences. Compared to the FID score, the KID score is more reliable by computing the squared Maximum Mean Discrepancy (MMD). For further evaluation of fidelity and diversity, we compute precision and recall. Additionally, the variance of generated motions is calculated as a diversity score. For motion representation comparison, we introduce a smoothness score to evaluate temporal features. More information can be found in the Appendix A.3.

## 5 RESULTS AND DISCUSSION

In this section, we break down the motion representation results into two sections, focusing on the qualitative and quantitative impacts of this particular decision. We also explore the performance between $v$MDM and state-of-the-art methods. To verify the validity of the modified loss function, we conduct an ablation study on loss functions. Training the motion diffusion model typically takes a few days on an A100 GPU. Therefore, we also compare the training time between $v$MDM and MDM.

### 5.1 QUANTITATIVE COMPARISONS

We calculate the values by using the metrics in Section 4.3 from two perspectives: representation comparisons and method comparisons. First, we dive into the impacts of various motion representations on $v$MDM and MDM. Second, we compare the evaluation performance of $v$MDM with the other three state-of-the-art methods. Following MDM, we sample 1000 generated motion sequences for all experiments to evaluate performance.

Table 2: Performance comparisons between $v$MDM and state-of-the-art methods based on HumanAct12 motion dataset.

| Method | FID↓ | KID↓ | Precision↑ | Recall↑ | Diversity↑ |
|---|---|---|---|---|---|
| ACTOR (6D) | 48.80 | 0.53 | 0.72 | 0.74 | 14.10 |
| MoDi (Quat.) | **13.03** | **0.12** | 0.71 | 0.81 | 17.57 |
| MDM (6D) | 31.92 | 0.36 | 0.66 | 0.62 | 17.00 |
| $v$MDM (JP) | 27.87 | 0.34 | **0.73** | **0.88** | **18.28** |

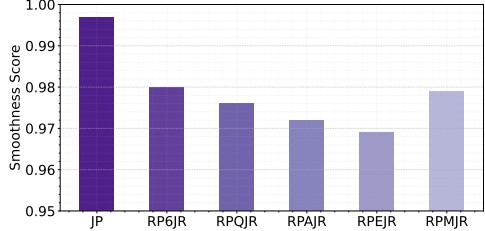

Figure 2: The average smoothness scores of six motion representations across the motion clips of the HumanAct12 motion dataset.

*Representation Comparisons.* We train 500K steps on $v$MDM and MDM under various motion representations based on the HumanAct12 motion dataset. Using JP motion representation on $v$MDM gets the best performance in all evaluation dimensions. At the same time, JP motion representation on MDM also outperforms in recall and diversity. In the experiments on $v$MDM, rotation-based motion representations perform poorly compared to MDM. For RPMJR rotation representation, we observe that $v$MDM is overfitting within 500K steps based on the change of the training loss. For effective comparisons, we sample the generated motion sequences for RPMJR representation from the $v$MDM that is trained for 300K

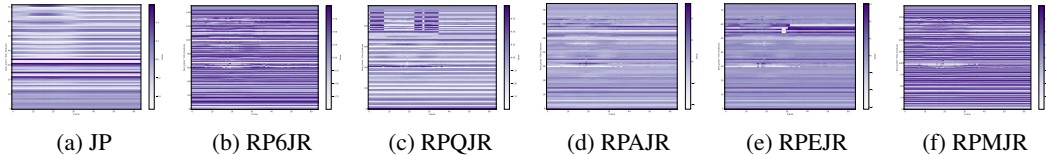

| (a) JP | (b) RP6JR | (c) RPQJR | (d) RPAJR | (e) RPEJR | (f) RPMJR |

Figure 3: Feature heatmaps of a single motion clip with diverse motion representations based on the HumanAct12 dataset.

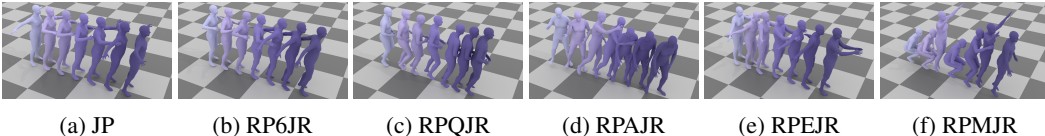

| (a) JP | (b) RP6JR | (c) RPQJR | (d) RPAJR | (e) RPEJR | (f) RPMJR |

Figure 4: Qualitative comparisons of generated motion sequences with various motion representations from $v$MDM based on HumanAct12 motion dataset. For clear visualization, we separate the poses at equal intervals across the frames (10-frame interval). The lighter the color, the smaller the frame index, and vice versa.

steps. In contrast to the $v$MDM, using RP6JR and RPMJR motion representations on MDM performs well on FID and KID indicators. $v$MDM is more capable of learning the latent distribution from a position-based motion representation in contrast to a rotation-based motion representation. For further comparisons of various motion representations, we calculate the average smoothness scores of the temporal features across all motion clips in the HumanAct12 motion dataset with the six motion representations. As shown in Figure 2, the smoothness score is the highest for the JP motion representation in the HumanAct12 motion dataset. For RP6JR and RPMJR motion representations, the smoothness scores are also higher than other three rotation representations. Either in MDM or $v$MDM, we suggest that the quality and diversity of motion generation are relevant to the smoothness of temporal features. We also visualize the feature heatmaps of a single motion clip as shown in the Figure 3. Compared to position-based motion representations, all rotation-based motion representations contain noisy points. Therefore, we assume that these noisy points are the primary cause of artifacts.

*Method Comparisons.* For the three state-of-the-art methods (further details are provided in the Appendix A.7), the three methods all use rotation-based motion representations to train the model. MoDi has the best performance on FID and KID indicators using a quaternion-based motion representation. However, quaternion rotation representation is challenging to grasp in the diffusion model due to the discontinuity of the quaternion. Therefore, the 6D rotation representation with continuity is more commonly used in the diffusion model for human motion synthesis. As shown in Table 2, both ACTOR and MDM utilize joint rotations in a 6D rotation representation as part of their motion representation, achieving competitive performance. Compared to rotation-based representation, $v$MDM with position-based motion representation (JP) demonstrates strong quantitative performance in precision, recall, and diversity evaluation indicators.

## 5.2 QUALITATIVE COMPARISONS

We visualize some generation results based on $v$MDM using HumanAct12 motion dataset with various motion representations as shown in Figure 4. Since the global movements of the generated motions are relatively small, we demonstrate the poses at equal intervals. By using JP motion representation, the generated poses are dynamic and diverse, but sometimes with keyframe popping. Compared to the position-based representation, the joint rotations are more seamless, but with more frequent frozen frames. For RP6JR motion representation, the generated pose switching is smooth, but floating still exists. For RPQJR motion representation, the inconsistency between global displacements and local poses is obvious, and it is prone to drifting. For RPAJR motion representation, motion freezing occurs in the motion sequence. For RPEJR motion representation, most generated motions are with smooth rotation continuity, but still with invalid rotations. Under the same hyperparameter setting, we observe that the convergence speed of the training process using RPMJR motion representation is extremely slow. The motion sequence generated by $v$MDM with RPMJR motion representation is continuous, but with motion floating. We visualize the feature heatmaps with various motion representations, as shown in Appendix A.8, for further analysis.

### 5.3 ABLATION STUDY

The loss function is crucial in the neural net-work training for human motion generation. In this section, we explore the influence of a simple loss function (only using the $v$ loss function) and a complex loss function (using the combination of the $v$ loss function and the geometric loss function) on $v$MDM. Empirically, we apply the Gaussian filter to fix the jittering problem of generated motion sequences. We also conduct an ablation study on motion temporal smoothing in Appendix A.6.

Table 3: Performance comparisons on the Human-Act12 dataset using $v$MDM with and without geometric loss.

| Loss | FID↓ | KID↓ | Precision↑ | Recall↑ | Diversity↑ |
|---|---|---|---|---|---|
| w/o $\mathcal{L}_g$ | 42.86 | 0.69 | **0.75** | 0.84 | **18.64** |
| w/ $\mathcal{L}_g$ | **27.87** | **0.34** | 0.73 | **0.88** | 18.28 |

For ablation evaluation about loss function design, we compare the evaluation performance based on $v$MDM with and without geometric loss. As shown in the Table 3, the precision and diversity scores are higher when the geometric loss is removed. Notably, $v$MDM, applying a complete loss function (i.e., with geometric loss), shows a significant improvement across FID, KID, and recall indicators.

### 5.4 COMPUTATIONAL COST COMPARISONS

For researchers, training on the motion diffusion model for human motion generation tends to require a large amount of time. Faster neural network training can reduce the computational resources and enable more efficient applications.

Table 4: Comparisons of training duration based on three methods with various motion representations on the HumanAct12 dataset.

| Method | JP | RP6JR | RPQJR | RPAJR | RPEJR | RPMJR |
|---|---|---|---|---|---|---|
| MDM | ∼1d | ∼3d | ∼3d | ∼3d | ∼3d | ∼3d |
| $v$MDM (w/o $\mathcal{L}_g$) | ∼**7h** | ∼**8h** | ∼**8h** | ∼**7h** | ∼**7h** | ∼**8h** |
| $v$MDM (w/ $\mathcal{L}_g$) | ∼1d | ∼3d | ∼3d | ∼3d | ∼3d | ∼3d |

The training duration is quite long while training the neural network following MDM. Therefore, we hope to find a way to keep a balance between training efficiency and motion generation quality. When we only apply the $v$ loss function, the training speed can be significantly enhanced. For $v$MDM with geometric loss and MDM, the training time can be reduced by two-thirds while using JP as motion representation. Therefore, we believe that a large amount of forward kinematic calculation using rotation-based motion representation sharply increases the model throughput. Suppose we aim to further accelerate the training process and slightly lower our expectations for the performance of motion generation. In that case, we can only apply the $v$ loss function to the motion diffusion model.

## 6 CONCLUSION

Our work demonstrates the impacts of various motion representations on the motion diffusion model. Quantitative comparisons of human motion generation performance reveal that $v$MDM using JP motion representation is better at handling position-based motion representation. Utilizing the JP motion representation can also accelerate the neural network training process by reducing the overhead for forward kinematics. In terms of qualitative analysis, the motions generated by $v$MDM with position-based motion representation (JP) are coherent and natural despite the potential for morphology changes. By contrast, the motions generated with RP6JR motion representation are smoother than those generated by other rotation-based motion representations due to the high continuity of temporal features. Our study has some limitations. The generation quality of $v$MDM is not outstanding under rotation-based motion representation without additional data cleaning. Temporal smoothing is necessary to address issues of jittering or frame popping. Determining whether the motion sequences generated perform best in terms of motion dynamics remains challenging. We observe that motion popping occurs frequently with position-based motion representation, whereas motion freezing and drifting are common with rotation-based motion representation. In the future, we aim to improve motion quality while maintaining diversity in human motion generation. Furthermore, we intend to provide a more comprehensive analysis of motion data itself, measuring not only the distribution similarity between real and generated motions but also proposing additional evaluation metrics to assess motion quality and continuity. Additionally, we hope to dive into raw motion

modeling and latent-space modeling of complex motion datasets, including sudden discontinuities, high velocities, rapid root rotation, and other features.

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

## A APPENDIX

### A.1 MOTION REPRESENTATION

*Joint Positions (JP).* We calculate the 3D coordinates of all joints local to the root space. The root position of the initial pose is at the origin of the world coordinate system. This motion representation at $i$th frame can be denoted as: $x_i = [pos_i] \in \mathbb{R}^{J \times 3}$, where $J$ is the number of the joints.

*Root Positions and 6D Joint Rotations (RP6JR).* To synthesize the global transformation of human motion, we concatenate the 3D positions of the root joint with the rotations of all joints using various rotation representations. To align with the dimensions of rotation representation, we perform zero-padding on the root position vector. We first convert the axis-angle rotation representation (SMPL pose parameters) to the rotation matrix and then to the 6D rotation representation. This motion representation at $i$th frame can be denoted as: $x_i = [rot_i^{6d}, pos_i^{root}] \in \mathbb{R}^{(J+1) \times 6}$.

*Root Positions and Quaternion Joint Rotations (RPQJR).* Like 6D rotation conversion, we process our data to get a quaternion rotation representation. To address the duality and discontinuity issues in quaternion representation, we utilize QuaterNet to enable a smooth transition between consecutive frames (Pavllo et al., 2018). This motion representation at the $i$th frame can be denoted as: $x_i = [rot_i^q, pos_i^{root}] \in \mathbb{R}^{(J+1) \times 4}$.

*Root Positions and Axis-angle Joint Rotations (RPAJR).* The SMPL pose parameters are encoded by an axis-angle rotation representation. We concatenate the root positions with the pose parameters, and this motion representation at $i$th frame can be denoted as: $x_i = [rot_i^a, pos_i^{root}] \in \mathbb{R}^{(J+1) \times 3}$.

*Root Positions and Euler Joint Rotations (RPEJR).* Similar to 6D rotation conversion, we first convert the axis-angle rotation representation to the rotation matrix. The Euler rotation representation is derived from the rotation matrix and expressed in radians. This motion representation at $i$th frame can be denoted as: $x_i = [rot_i^e, pos_i^{root}] \in \mathbb{R}^{(J+1) \times 3}$.

*Root Positions and Matrix Joint Rotations (RPMJR).* We directly convert the axis-angle rotation representation to $3 \times 3$ rotation matrix representation. This motion representation at $i$th frame can be denoted as: $x_i = [rot_i^m, pos_i^{root}] \in \mathbb{R}^{(J+1) \times 9}$.

### A.2 DATASET

*HumanAct12 Dataset.* The HumanAct12 dataset (Guo et al., 2020) consists of 1,191 motion clips, including 12 coarse action categories: warm up, walk, run, jump, drink, lift dumbbell, sit, eat, turn steering wheel, phone, boxing, and throw. The total number of frames is 90,099 (20 fps, around 1.3 hours). This dataset is based on the SMPL skeleton (24 joints, 23 bones).

*100STYLE Dataset.* The 100STYLE dataset (Mason et al., 2022) features over 4 million frames of motion capture data, showcasing 100 distinct locomotion styles. In this paper, we focus on forward walking and running because forward locomotion skills are more commonly applied in a large number of scenarios. Therefore, our experiments are conducted based on the forward locomotion motion dataset. The number of total frames in the forward locomotion dataset is 596,429 (around 5.5 hours). We remove the unnecessary poses by setting the indices of the beginning frame and ending frame of each motion clip. Next, we downsample the motion data from 60 fps to 30 fps, and we retarget all motion sequences to the SMPL skeleton. We split the overall motion data into multiple motion sequences within 64 frames (around 2 seconds). Our motion dataset from 100STYLE contains 9,228 motion sequences.

*HumanML3D Dataset.* The HumanML3D dataset (Guo et al., 2022a) originally consists of 14,616 motion clips. This dataset contains a wide range of action categories, not limited to locomotion skills. In our training dataset, we have 29,224 motion clips with 64 frames each (20 fps, totaling around 26.0 hours), split from long motion clips. We also follow the 22-joint SMPL skeleton structure.

### A.3 EVALUATION METRICS

*Fréchet Inception Distance (FID).* FID is a universal metric used to measure the quality and diversity of the generated data. A lower FID score signifies a closer alignment between real and generated data distribution. Similar to image generation, an action recognition model from (Raab et al., 2023) is used to encode the generated motion sequences based on HumanAct12 dataset into the latent feature vectors. Let the distribution of the real motion features be denoted by $\mathcal{N}(\mu_r, \Sigma_r)$ and the distribution of the generated motion features be denoted by $\mathcal{N}(\mu_g, \Sigma_g)$. The FID score can be computed by:

$$FID = \|\mu_r - \mu_g\|_2^2 + tr(\Sigma_r + \Sigma_g - 2(\Sigma_r\Sigma_g)^{\frac{1}{2}}) \tag{13}$$

We sample 1,000 generated motion sequences from the diffusion models based on six motion representations. Then we compute FID scores between the generated motion samples with various motion representations and real motion samples. The lower the FID score, the better the motion generation performance.

*Kernel Inception Distance (KID).* KID is a metric that measures the difference between real and generated data features using Maximum Mean Discrepancy (MMD). A lower KID score indicates a higher quality and diversity of the generated motion data. The computation of KID score is as follows:

$$KID = MMD^2(f_r, f_g) \tag{14}$$

where $f_r$ and $f_g$ are the features computed from the action recognition network separately.

*Precision and Recall.* Precision is an indicator of fidelity measurement by computing the probability that a generated motion is contained within the distribution of real motion using the k-th nearest neighbour (kNN) distance. Similarly, recall is an indicator of diversity measurement, computed by determining the probability that a real motion is contained within the distribution of generated motions using the kNN distance. Higher precision and recall values indicate higher fidelity and diversity in the generated motion.

*Diversity.* This metric is adapted from Guo et al. (2020). Diversity measures the variance of the generated motions over the full set of action categories. Two subsets are sampled from the features of generated motions and real motions. The size of the two subsets is the same. Diversity score is computed by:

$$Diversity = \frac{1}{S_d} \sum_{i=1}^{S_d} \left\| \mathbf{f}_r^i - \mathbf{f}_g^i \right\|_2 \tag{15}$$

where $s_d = 200$ and the data of the two subsets are both randomly sampled.

*Smoothness.* This metric is based on the first and second derivatives of the temporal data. For our motion dataset, we use this metric to evaluate the continuity of all features over time. If the motion clips are commonly smooth and natural, the second derivatives are small, and thus the score approaches 1. The calculation of smoothness is as follows:

$$Smoothness = \frac{1}{N \times M} \sum_{i=1}^{N} \sum_{j=1}^{M} \left( \frac{1}{T-2} \sum_{t=1}^{T-2} e^{-\alpha |a_{(i,j),t}|} \right) \tag{16}$$

where $N$ is the number of motion clips in the motion dataset, $M$ is the number of feature dimensions, $T$ is the number of frames for each motion clip, $\alpha$ is the sensitivity parameter and is more than 0, $a_{(i,j),t}$ is the discrete acceleration of feature $j$ of motion clip $i$ at time $t$.

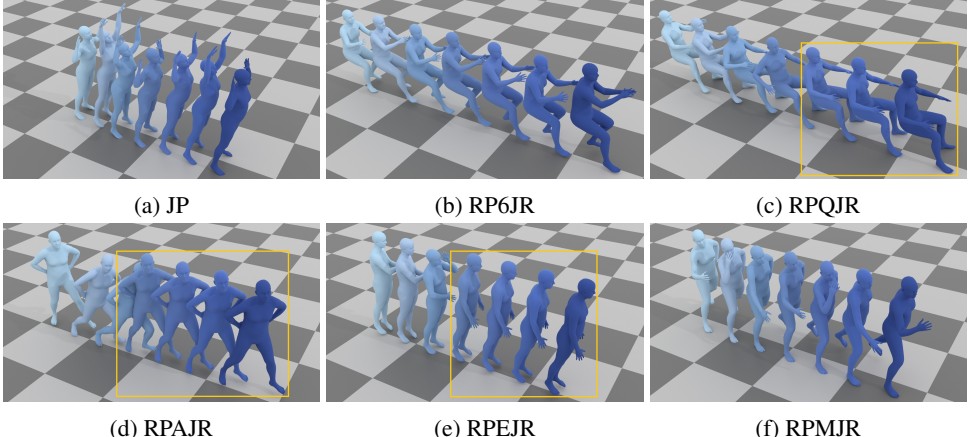

(a) JP  (b) RP6JR  (c) RPQJR

(d) RPAJR  (e) RPEJR  (f) RPMJR

Figure 5: Qualitative comparisons of generated motion sequences with various motion representations from MDM based on HumanAct12 motion dataset. For clear visualization, we separate the poses at equal intervals across the frames. The lighter the colour, the smaller the frame index, and vice versa. In the yellow box, the poses remain almost stationary.

## A.4  GENERATED MOTIONS USING MDM

We also generate motion sequences from MDM with various motion representations, and the visualization results are as shown in Figure 5. For JP, RP6JR, and RPMJR motion representations, each single pose in the motion clip is dynamic and continuous. As opposed to these motion representations, motion freezing occurs in the generated motion samples with RPQJR, RPAJR, and RPEJR motion representations. For $v$MDM and MDM, using JP motion representation can both generate high-quality motions.

## A.5  ROBUSTNESS EVALUATION

To test the robustness of $v$MDM, we conduct training with JP motion representation based on the 100STYLE dataset that contains extensive motion ranges. The visualization results are shown in Figure 6. Based on the large-scale motion dataset, $v$MDM can generate diverse styles and stable foot grounding. MDM can also generate high-quality motion sequences, but with slight foot sliding and motion drifting.



Figure 6: Qualitative results of generated motion sequences with JP motion representation from $v$MDM (purple) and MDM (blue) based on 100STYLE motion dataset. We visualize the poses across the frames (1st frame, 10th frame, 20th frame, 30th frame, 40th frame, 50th frame, and 60th frame).

## A.6  GAUSSIAN FILTER

As shown in Figure 7, we can intuitively see the differences between using a Gaussian filter and not using one. Without a Gaussian filter, the direct pelvis movements obtained from $v$MDM are jittering. A Gaussian filter can effectively improve this problem and is capable of making the generated motions more continuous and stable.

Jitter is commonly observed among the generated motion sequences based on diffusion models. However, we find that applying the $v$ loss function can lead to jitter as shown in the second row of

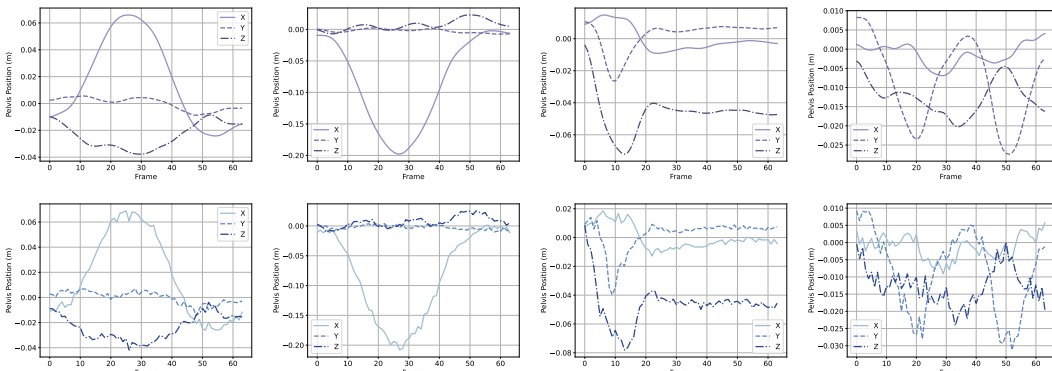

Figure 7: Ablation study of temporal smoothing. We sample four motion sequences generated by $v$MDM based on the HumanAct12 motion dataset and compare the pelvis position change across the frames with (first row, purple) and without (second row, blue) a Gaussian filter.

Figure 7. When the prediction objective of the diffusion model introduces noise-related terms, the generated results can also be influenced and thus contain noise. In this case, to achieve stable and clean generation results, additional loss constraints or post-processing are required. The introduction of more loss constraints may increase training costs to some extent.

## A.7  BASELINE METHODS

We compare our proposed method with three state-of-the-art methods, including ACTOR (Petrovich et al., 2021), MoDi (Raab et al., 2023), and MDM (Tevet et al., 2022b). At the same time, we dive into the impacts of our integrated motion representations by comparing MDM and $v$MDM.

*ACTOR.* Action-Conditioned TransfORmer VAE (ACTOR) is proposed in Petrovich et al. to generate the motion sequences based on the Transformer. ACTOR utilizes a combination of joint rotations, 6D rotation representation, and root positions as motion representations.

*MoDi.* MoDi is based on StyleGAN architecture to synthesize high-quality motion sequences from diverse data. By encoding the motion into the latent space, the diverse motions are generated with rich semantics. Modi uses joint rotations with quaternion rotation representation, root positions and velocities, and foot contact labels as motion representation.

*MDM.* Motion Diffusion Model (MDM) is proposed by Tevet et al. for multi-task motion generation based on diffusion. We use the HumanAct12 motion dataset for evaluation; therefore, we focus on the motion representation employed in this dataset. The motion representation is the combination of joint rotations in the 6D rotation representation and root positions.

## A.8  FEATURE HEATMAP

For intuitive comparison of different feature maps, we visualize the feature heatmaps with diverse motion representations based on HumanAct12 and 100STYLE datasets.

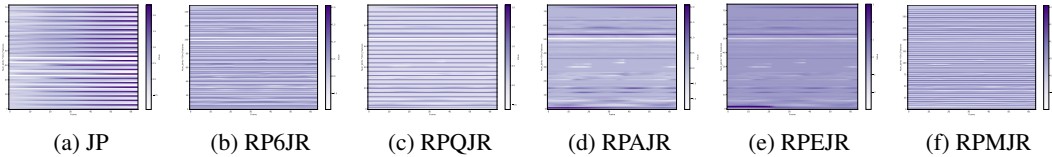

|  (a) JP | (b) RP6JR | (c) RPQJR | (d) RPAJR | (e) RPEJR | (f) RPMJR |

Figure 8: The feature heatmaps of a single motion clip with diverse motion representations based on the 100STYLE dataset.

By visualizing the feature heatmaps of motion clips from two motion datasets (as shown in Figure 3 and Figure 8), the 100STYLE motion dataset is found to have higher quality. Our qualitative

results indicate that the quality of the motion dataset matters for motion generation quality. For rotation-based motion representation, the feature heatmaps for the HumanAct12 motion dataset have more noisy points. Although MDM with RP6JR motion representation shows a strong capacity for human motion generation, the generated motions still contain some artifacts. We suppose this is relevant to the training motion data quality and motion representation quality. In generated motion sequences, more artifacts (e.g., drifting, foot sliding, keyframe popping, motion freezing, invalid or unnatural pose, etc.) are more regularly encountered in the model based on the HumanAct12 dataset compared to the 100STYLE dataset. Based on the better performance on $v$MDM with JP motion representation, we believe that $v$MDM is better at handling feature information with a higher rate of change. However, the feature of a high rate of change and the loss function based on derivatives also lead to a notable artifact: jittering. A Gaussian filter is effective in eliminating the jittering issue, ensuring the generated motions are of high quality.

## A.9 RESULTS ON HUMANML3D DATASET

We also train $v$MDM on the HumanML3D motion dataset for comprehensive evaluation. As shown in the Figure 9, $v$MDM is still capable of generating smooth motions by using JP motion representation. These high-quality generated motions further demonstrate the effectiveness of $v$MDM with the JP motion representation on benchmark datasets.

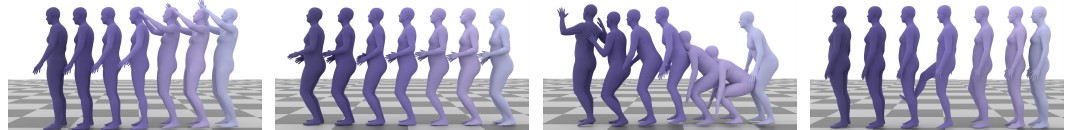

Figure 9: Qualitative results of generated motion sequences with JP motion representation from $v$MDM based on HumanML3D motion dataset. We visualize the poses at fixed intervals across the frames (1st frame, 10th frame, 20th frame, 30th frame, 40th frame, 50th frame, and 60th frame).

