# OpenReview forum: "Back to Basics: Motion Representation Matters for Human Motion Generation Using Diffusion Model"
_ICLR.cc/2026/Conference — Submitted to ICLR 2026_

### Official Review · Reviewer_Dy62 · 2025-10-31

**Soundness:** 3
**Presentation:** 3
**Contribution:** 3
**Rating:** 6
**Confidence:** 3

**Summary:**

This paper investigates the impact of motion representations and loss functions on human motion generation using diffusion models. It proposes vMDM, an extended version of the Motion Diffusion Model (MDM) that adopts the v loss (a weighted combination of motion data and noise) as the prediction objective, supplemented with geometric losses (position, velocity, foot contact loss). The authors systematically compare six common motion representations (one position-based: JP; five rotation-based: RP6JR, RPQJR, RPAJR, RPEJR, RPMJR) on two datasets (HumanAct12 and retargeted 100STYLE to SMPL skeleton). Key results show that the position-based JP representation outperforms rotation-based ones in vMDM in terms of diversity, fidelity, and training efficiency, while rotation-based representations (e.g., RP6JR) excel in motion stability. Additionally, the integration of v loss balances generation performance and training speed, and temporal smoothing with a Gaussian filter mitigates jitter issues.

**Strengths:**

1. Focuses on fundamental yet under-explored questions (motion representations and loss design) in diffusion-based human motion generation, filling the gap of inconsistent motion representation standards in existing work.

2. Rigorously evaluates six motion representations across quantitative (FID, KID, precision, recall, diversity, smoothness) and qualitative metrics, with sufficient ablation studies on loss functions and temporal smoothing.

3. The proposed vMDM (v loss + geometric loss) achieves better performance than baseline methods (ACTOR, MoDi, MDM) on key metrics, and JP representation reduces training time by avoiding complex forward kinematics computations.

4. Retargets the large-scale 100STYLE dataset to the SMPL skeleton, verifying the framework’s effectiveness across datasets with limited and extensive motion ranges.

**Weaknesses:**

1. vMDM performs poorly with rotation-based representations without additional data cleaning, and the paper fails to propose targeted improvements for this limitation.

2.  Position-based JP representation suffers from keyframe popping, while rotation-based ones face freezing/drifting; temporal smoothing (Gaussian filter) is a post-processing patch rather than a fundamental solution.

3. Only compares with three baseline methods, missing recent state-of-the-art diffusion-based motion generation models. Evaluation metrics lack assessments of motion dynamics and physical plausibility (e.g., balance, momentum conservation).

4. The weight setting of the v loss (ω(t)=ᾱₜ) and geometric loss components lacks detailed justification; the impact of hyperparameters (e.g., diffusion time steps, Transformer structure) on different motion representations is not discussed.

**Questions:**

1. What is the theoretical or empirical basis for choosing ω(t)=ᾱₜ as the weight of the v loss? Have you tested other weight functions (e.g., adaptive weights based on motion types) and their impacts?

2. Why do rotation-based representations introduce more noisy points (as shown in feature heatmaps)? Is it related to data preprocessing, representation inherent properties, or model architecture?

3. How does vMDM perform on datasets with more complex motions (e.g., acrobatics, interactive motions) beyond locomotion and simple actions? Can JP representation maintain its advantages in such scenarios?

---

> ### Author Response · Authors · 2025-11-24
>
> We sincerely thank the reviewer for the detailed and thorough reviews.
>
>
> ## Loss
> SNR weighting is widely used in diverse diffusion models. The SNR level measures the level of noise in the motion data. Based on [1], the loss weighting strategy effectively balances different timesteps. For $v$ prediction, it is a combination of $x_0$ and $\epsilon$. By adding SNR weights to the loss function, the $v$-prediction loss is equivalent to the noise-prediction loss mathematically.
>
>
> $\mathcal{L}_{\epsilon}$
>
> $= \| \epsilon_t - \hat{\epsilon}_\theta \|_2^2$
>
> $= \| (\alpha_t v_t + \sigma_t x_t) - (\alpha_t \hat{v}_\theta + \sigma_t x_t) \|_2^2$
>
> $= \| \alpha_t (v_t - \hat{v}_\theta) \|_2^2$
>
> $= \alpha_t^2 \| v_t - \hat{v}_\theta \|_2^2$
>
> $= \frac{\alpha_t^2}{\sigma_t^2} \cdot \frac{\sigma_t^2}{\alpha_t^2 + \sigma_t^2} \| v_t - \hat{v}_\theta \|_2^2$
>
> $= \frac{\mathrm{SNR}}{\mathrm{SNR} + 1}  \| v_t - \hat{v}_\theta \|_2^2$
>
> $\omega(t) = \bar{\alpha}_t = \frac{\mathrm{SNR}}{\mathrm{SNR} + 1}$
>
>
>
>
> where $\alpha_t=\sqrt{\bar{\alpha_t}}$, $\sigma_t = \sqrt{1 - \bar{\alpha_t}}$, $\alpha_t^2+\sigma_t^2=1$, $SNR=\frac{\alpha_t^2}{\sigma_t^2}$.
>
> Applying adaptive weights given the motion type is a potential way to improve model learning, and we will test it in our future work.
>
>
> ## Rotation Representation
> It is relevant to the quality of the motion dataset and the rotation properties. Clean, continuous joint rotation changes require a high-quality motion dataset. Compared to HumanAct12 motion dataset, 100STYLE dataset has better continuity in motion representation heatmaps. When we visualize the motions from two motion datasets, we observe that 100STYLE motion dataset is cleaner and more stable. If the training data contains artifacts, the position-based motion representation is cleaner than the rotation-based motion representation in terms of continuity. For short-term human motion generation, the range of joint position changes is not much bigger. But joint rotations can always be scaled to a fixed range.
>
> ## Generalization
> So far, we focus on the learning of basic motion and locomotion skills. In CAMDM [2], they focus only on locomotion learning using a diffusion model for character control. In the future, we will first make a comprehensive analysis of diverse motion datasets. Then we will dive into multiple-category motions, including sudden discontinuities, high velocities, or rapid root rotation, etc, and explore the impacts of motion representation on complex motions.
>
> ## References
> [1] Efficient diffusion training via min-snr weighting strategy.
>
> [2] Taming diffusion probabilistic models for character control.

---

### Official Review · Reviewer_j864 · 2025-10-31

**Soundness:** 2
**Presentation:** 2
**Contribution:** 2
**Rating:** 4
**Confidence:** 5

**Summary:**

In this paper, the authors revisit core design choices for diffusion-based human motion generation, examining how motion representation and objective choice affect quality, diversity, and efficiency. Building on MDM, they adopt a v-parameterized objective within a controlled architecture and systematically compare six representations—joint positions and five rotation parameterizations—under a unified training/evaluation setup on HumanAct12 and a 100STYLE-to-SMPL benchmark. The main result is that JP with vMDM achieves the best overall FID/KID and recall, while rotation features yield smoother transitions but more artifacts. The study further includes ablations on representation choice, loss design, and compute, showing notable training-time gains for vMDM and additional speedups with JP by avoiding forward kinematics.

**Strengths:**

1. The manuscript tackles the impact of motion representation on diffusion-based motion generation—a meaningful question that can inform and inspire subsequent research.
2. The manuscript provides thorough quantitative and qualitative analyses across multiple motion representations, offering clear empirical evidence to support the study’s conclusions.

**Weaknesses:**

1. **Limited methodological breadth**. Experiments are confined to MDM, without evaluating other motion-generation methods (e.g., VAE- or autoregressive-based models, or more recent architectures). This narrow scope limits the generality and external validity of the conclusions.
2. **No study of combined representations**. Each representation is trained and tested in isolation. As noted by the authors, prior work and datasets (e.g., HumanML3D [1]) often combine permutations of joint positions and joint rotations to exploit complementary cues. Without exploring combined representations, the study cannot assess potential synergies and its practical meaning is reduced.
3. **Dataset scope is narrow**. Results are reported only on 100STYLE and HumanAct12, omitting broader and widely used benchmarks such as AMASS [2] and the HumanML3D [1] used by MDM. This limits coverage, motion diversity, comparability to prior work, and cross-dataset generalization, weakening the strength of the empirical claims.

> [1] Guo, Chuan, et al. "Generating diverse and natural 3d human motions from text." *Proceedings of the IEEE/CVF conference on computer vision and pattern recognition*. 2022.

> [2] Mahmood, Naureen, et al. "AMASS: Archive of motion capture as surface shapes." *Proceedings of the IEEE/CVF international conference on computer vision*. 2019.

**Questions:**

The results in this manuscript report that joint positions produce more coherent and natural motion. However, many downstream applications require SMPL parameters, and converting JP to SMPL adds a post-processing/IK step that is time-consuming and can introduce rotation errors. Prior work (e.g., MotionStreamer [3]) therefore combines joint positions and rotations. How do you view this trade-off in practice when choosing motion representations.

> [3] Xiao, Lixing, et al. "MotionStreamer: Streaming Motion Generation via Diffusion-based Autoregressive Model in Causal Latent Space." *arXiv preprint arXiv:2503.15451*(2025).

---

> ### Author Response · Authors · 2025-11-24
>
> We sincerely thank the reviewer for the detailed and thorough reviews.
>
> Yes, this is a classical problem, and we strongly agree that it is up to the downstream tasks. Especially in animation or gaming, joint rotations are essential for mesh rigging or motion retargeting. For general human motion generation, position-based motion remains useful across multiple areas, including motion generation, synthesis, retrieval, and planning. We've taken steps to clarify the nuance in the outcomes of our work.

---

> ### Author Response · Authors · 2025-12-03
>
> We also train $v$MDM on the HumanML3D motion dataset for comprehensive evaluation and still generate high-quality motions by using joint positions as motion representation. Relevant results can be seen in Appendix A.9 and the supplementary video (HumanML3D starting at around 3'08'').

---

### Official Review · Reviewer_Dxar · 2025-10-31

**Soundness:** 2
**Presentation:** 2
**Contribution:** 2
**Rating:** 2
**Confidence:** 4

**Summary:**

This paper studies fundamental in human motion generation, the representation itself. The authors take the MDM architecture and train various alternatives with different representations and with x0 and v prediction. In specific, they compare six motion representations — joint positions,6D rotations + root, quaternion + root, axis-angle + root, euler + root, rotation matrices + root. Experiments are conducted on two different datasets, HumanACT12 and 100STYLE, using quantitative metrics common in the motion generation field (FID, R-precision, diversity, etc). Overall, the authors argue that representation JP and v-loss surpass other qualities, stability, and efficiency in motion diffusion.

**Strengths:**

1. Modern advancements in the motion generation field are mostly at the architectural level; however, representation itself is the fundamental problem as well as the diffusion objective. As a researcher in this field, I agree with the author's motivation for the paper.
2. Various comparisons across six representations are conducted on a controlled architecture, and the results follow intuition and motivation.

**Weaknesses:**

1. Though I admire the motivation and respect the authors’ effort to explore representation-level questions in motion generation. As a researcher in this field, **I sincerely believe it is also important to acknowledge prior work that has already addressed many of these issues**. Several earlier papers have studied these choices and reported similar or stronger findings before this submission.

MLD, MotionLCM, and MotionStreamer have shown that latent spaces are a much better representation to model than raw motion representations (for both x0 and noise prediction).

SALAD, MARDM, and  ACMDM have shown that v prediction is better than x0 prediction and noise prediction in both latent and raw motion data.

ACMDM shows that absolute joint positions are better than other representations for various motion generation tasks, even with the simplest architecture.

SALAD and ACMDM show that condition injection is also an important factor in the generation quality.

And above method also conducts experiments in the standard benchmark in HumanML3D and KIT-ML datasets.

2. There are now many larger, more diverse benchmarks for human motion generation, and HumanML3D in particular has long became the default dataset for evaluating text-conditioned motion models in recent work. Without results on a standard benchmark like HumanML3D, the general validity of the experimental conclusions in this paper is difficult to assess. Especially as MDM is considered a quite old method, with many more methods that surpass MDM.

3. The paper claims to retarget 100STYLE for evaluation; however, prior work (e.g., SMooDi) has already performed 100STYLE retargeting to common skeleton formats more than a year ago.

4. While representation choice and the use of v-prediction are important, motion generation quality is also strongly affected by how conditioning is injected. This is left out in the paper, while previous works such as SALAD, ACMDM have proven that cross attention and AdaLN are much better than in-context learning used in MDM.

5. The visual results of the overall best method: v prediction and JP still exhibit heavy floating and jittering, shown in the supplemental video.

[HumanML3D]Guo, Chuan, et al. "Generating Diverse and Natural 3D Human Motions from Text." CVPR 2022

[KIT-ML]Plappert, Matthias, et al. "The kit motion-language dataset." Big data 2016.

[MLD]Chen, Xin, et al. "Executing your commands via motion diffusion in latent space." CVPR 2023.

[MotionLCM]Dai, Wenxun, et al. "Motionlcm: Real-time controllable motion generation via latent consistency model." ECCV 2024

[MotionStreamer]Xiao, Lixing, et al. "MotionStreamer: Streaming Motion Generation via Diffusion-based Autoregressive Model in Causal Latent Space." ICCV 2025.

[SALAD]Hong, Seokhyeon, et al. "SALAD: Skeleton-aware Latent Diffusion for Text-driven Motion Generation and Editing." CVPR 2025

[MARDM]Meng, Zichong, et al. "Rethinking Diffusion for Text-Driven Human Motion Generation." CVPR 2025.

[ACMDM]Meng, Zichong, et al. "Absolute Coordinates Make Motion Generation Easy." ArXiv 2025

[SmooDi]Zhong, Lei, et al. "Smoodi: Stylized motion diffusion model." ECCV 2024

**Questions:**

1. Why are FIDs so high in all tables in the experiment section? May I ask what the ground truth FID is? The high level of FID seems unreasonable and may lose the meaning of comparison.
2. Many work (MLD, MotionLCM, MotionLCM v2, MARDM, MotionStreamer, ACMDM) show that latent space modeling may potentially have different results than raw motion modeling, and is quickly becoming the default choice. Would the author also include a latent space comparison to make the experiment complete?

---

> ### Author Response · Authors · 2025-11-24
>
> We sincerely thank the reviewer for the detailed and thorough reviews.
>
> ## FID
> HumanAct12 is a small-scale dataset; its diversity is lower than that of the HumanML3D dataset. FID is the Fréchet distance between the Gaussian statistics of features from real vs generated motion. In MDM [1], the feature extraction approaches for HumanAct12 and HumanML3D are different. The ground truth FID score is 0. As shown in Table 5 of MDM, the FID scores are for HumanAct12. For each dataset, the level of FID scores are more critical to the evaluation results. For the same motion dataset, "high" and "low" FID scores indicate evaluation performance. Some works also use a scaling factor to calculate average FID scores across feature dimensions.
>
>
>
> ## Latent Space Comparison
> We appreciate this thoughtful suggestion. Our work aims to conduct empirical studies on raw motion modeling. We believe that latent space modeling has great potential on human motion generation. However, training a motion encoder and decoder to extract high-level features can still lead to motion reconstruction error. We consider latent space encoding in human motion generation an excellent future avenue of inquiry.
>
>
> ## References
> [1] Human motion diffusion model.

---

> ### Author Response · Authors · 2025-12-03
>
> We also train $v$MDM on the HumanML3D motion dataset for comprehensive evaluation and still generate high-quality motions by using joint positions as motion representation. Relevant results can be seen in Appendix A.9 and the supplementary video (HumanML3D starting at around 3'08'').

---

### Official Review · Reviewer_PvFL · 2025-11-01

**Soundness:** 2
**Presentation:** 3
**Contribution:** 2
**Rating:** 2
**Confidence:** 4

**Summary:**

This paper revisits motion representation for diffusion-based motion generation: starting from MDM/vMDM, they plug in six encodings (JP, RP6JR, RPQJR, RPAJR, RPEJR, RPMJR) and run a series of comparisons on HumanAct12 and 100STYLE.

**Strengths:**

1. Clean ablation of six motion representations within one diffusion backbone (MDM/vMDM) with standard metrics (FID/KID/precision/recall/diversity).

2. Clear empirical takeaway in their setup: under vMDM, JP outperforms rotation-based reps and is faster to train.

3. Practical notes on training/inference that practitioners can immediately try.

**Weaknesses:**

1. The general question “which motion representation is easier to learn for diffusion models” is not new. For example, MARDM [1] discusses redundant motion representations for training VQ-based vs. diffusion-based models. MotionStreamer [2] proposes a 272-D motion representation to remove post-processing that is required for animation. InterGen [3] introduces a representation tailored for two-person interactions. ACMDM [4] shows that absolute/global joint coordinates improve motion fidelity and text alignment, which is basically the same finding here (JP > others in diversity/fidelity/efficiency). Compared to this thread, the paper is incremental, it should explicitly acknowledge these works and clarify how its findings differ or inspire new directions.

2. All experiments are on HumanAct12 and 100STYLE. These are small-scale. I don’t see why the authors didn’t evaluate on widely used benchmarks like HumanML3D or KIT-ML, or newer/larger datasets such as SnapMoGen. Relying on small datasets makes the analysis less convincing.

3. No statistical variation (mean ± std across seeds) is reported.

4. As a benchmark/analysis paper, even though results are given, I still don’t know why one representation is better than another. The paper should add more experiments and analysis that can explain the reasons.

References:

[1] Rethinking Diffusion for Text-Driven Human Motion Generation

[2] MotionStreamer: Streaming Motion Generation via Diffusion-based Autoregressive Model in Causal Latent Space

[3] InterGen: Diffusion-based Multi-human Motion Generation under Complex Interactions

[4] Absolute Coordinates Make Motion Generation Easy

**Questions:**

Refer to Weaknesses.

---

> ### Author Response · Authors · 2025-11-24
>
> We sincerely thank the reviewer for the detailed and thorough reviews.
>
> ## Motion Representation
> We agree that similar important questions have been asked in the literature and we haev made this clear in the paper. We believe exploring which motion representation matters for diffusion approach is still meaningful. Joint position and rotation features are critical for human motion synthesis. In motion generation, MoDi [1] uses quaternion rotations to achieve high performance with a GAN. For diffusion learning, the body of literature shows a preference for 6D rotations, but our work shows this assumption may be more nuanced and that other representations have merit.
>
> In MARDM [2], they aim to explore two motion representations with/without redundant features:  ${x}_i = [ \dot{r}_a,\ \dot{r}_x,\ \dot{r}_z, \ \dot{r}_h,\ j_p ]$ and ${x}_i = [ \dot{r}_a,\ \dot{r}_x,\ \dot{r}_z,\ \dot{r}_h,\ j_p,\ j_v,\ j_r,\ c_f ]$.
>
> In MotionStreamer [3], this work makes a slight modification for motion representation: ${x}_i = [ \dot{r}_x,\ \dot{r}_z,\ \dot{r}_a,\ j_p,\ j_v,\ j_r ]$
> where we project the root on the XZ-plane (ground plane),
> $\dot{r}_x, \dot{r}_z \in \mathbb{R}$ are the root linear velocities on the XZ-plane,
> $\dot{r}_a \in \mathbb{R}^6$ denotes the root angular velocity represented in 6D rotations,
> $j_p \in \mathbb{R}^{3K}$, $j_v \in \mathbb{R}^{3K}$, and $j_r \in \mathbb{R}^{6K}$ are the local joint positions, local velocities, and local rotations relative to the root space, where $K$ is the number of joints.
>
> In InterGen [4], the motion representation is formulated as: ${x}_i = [\, j_p^{g},\ j_v^{g},\ j_r,\ c_f \,]$
> where the $i$-th motion state $\mathbf{x}_i$ is defined as a collection
> of global joint positions $j_p^{g} \in \mathbb{R}^{3N_j}$, velocities
> $j_v^{g} \in \mathbb{R}^{3N_j}$ in the world frame, 6D representations
> of local rotations $j_r \in \mathbb{R}^{6N_j}$ in the root frame, and binary
> foot--ground contact features $c_f \in \mathbb{R}^4$.
>
> In ACMDM [5], one uses absolute coordinates with kinematic-aware and redundant representation (InterGen), while the other uses plain absolute coordinates (ACMDM) (i.e., ${x}_i = [\, j_p^{g},\ j_v^{g},\ j_r,\ c_f \,]$ v.s. ${x}_i = [\, j_p^{g} \,]$).
>
>
>
>
> In summary, the above research is centered on the motion representation for HumanML3D. However, we aim to explore the impact of diffusion model learning with \textbf{position-based} $(x_i = [pos_i] \in \mathbb{R}^{J\times3})$ and \textbf{rotation-based representations} $x_i = [rot_i^{6d}, pos_i^{root}] \in \mathbb{R}^{(J+1)\times6}$, $x_i = [rot_i^{q}, pos_i^{root}] \in \mathbb{R}^{(J+1)\times4}$, $x_i = [rot_i^{a}, pos_i^{root}] \in \mathbb{R}^{(J+1)\times3}$, $x_i = [rot_i^{e}, pos_i^{root}] \in \mathbb{R}^{(J+1)\times3}$, $x_i = [rot_i^{m}, pos_i^{root}] \in \mathbb{R}^{(J+1)\times9}$. These motion representations are more common not only for human motion generation but also for human motion prediction, character control, etc.
>
> ## Dataset Selection
> Our focus was on fine locomotion skill learning for future research in that area. Our 100STYLE motion dataset contains 9,228 motion clips, which is $>$ 2x the size of KIT-ML. We feel that for the question at hand, these standard datasets were of appropriate size and diversity. We leave current, recently released datasets with text annotations to future work on conditional diffusion. We've added clarifications and references to that data in the paper.
>
> ## Statistical Variation
> We follow the evaluation of unconstrained synthesis on the HumanAct12 dataset in MDM [6]. The seed is fixed for unconditional human motion generation.
>
>
>
> ## Result
> We have now clarified this in the paper by highlighting the mapping between our outcomes and specific dataset outcomes. Based on our results, position-based motion representation outperforms in feature processing, training efficiency, motion generation diversity, and motion integrity. The 6D rotation motion representation outperforms other motion representations and achieves strong motion continuity. For position-based motion representation, the same feature semantics (root displacements and non-root joint positions) are essential for diffusion learning. However, there is a different semantics in rotation-based motion representation: root displacements and joint rotations.
>
> ## References
>
> [1] Modi: Unconditional motion synthesis from diverse data.
>
> [2] Rethinking
> diffusion for text-driven human motion generation: Redundant representations, evaluation,
> and masked autoregression.
>
> [3] Motionstreamer: Streaming motion generation via diffusion-based autoregressive model in causal latent space.
>
> [4] Intergen: Diffusion-based multi-human motion generation under complex interactions.
>
> [5] Absolute coordinates make motion generation easy.
>
> [6] Human motion diffusion model.

---

> ### Author Response · Authors · 2025-12-03
>
> We also train $v$MDM on the HumanML3D motion dataset for comprehensive evaluation and still generate high-quality motions by using joint positions as motion representation. Relevant results can be seen in Appendix A.9 and the supplementary video (HumanML3D starting at around 3'08'').

---

### Comment · Area_Chair_DCzb · 2025-11-25

Dear reviewers,

The authors have responded. We kindly ask you to review the authors' responses to your comments and provide your feedback. Thank you.

Best,

AC

---

### Meta-Review · Area_Chair_ufBV · 2025-12-24

**Summary:**

This paper investigates the impact of different motion representations (specifically comparing Joint Positions vs. various rotation-based parameterizations) and prediction targets (v-prediction vs. noise/x0) within a diffusion-based human motion generation framework (based on MDM). The authors propose vMDM and claim that position-based representations (JP) outperform rotation-based ones in terms of diversity, fidelity, and training efficiency in this specific setup.

The reviewers acknowledged that revisiting fundamental design choices like representation and loss objectives is a meaningful scientific question (PvFL, Dxar, j864, Dy62), also recognizing the clean ablation study and findings (PvFL, Dy62).

However, some major concerns are also raised:
1. Multiple reviewers pointed out that the question of representation has been explored in recent literature (e.g., MARDM, MotionStreamer, ACMDM, InterGen), often with conclusions favoring latent spaces or absolute coordinates which this paper does not fully address or benchmark against (PvFL, Dxar).
2. The study is limited to raw motion diffusion (MDM), ignoring the current state-of-the-art trend towards latent diffusion models (MLD, MotionLCM), which generally produce better results. Reviewers felt comparing only against older baselines limits the validity of the conclusions (Dxar, j864).
3. The initial submission relied on HumanAct12 and 100STYLE, omitting the standard HumanML3D benchmark, which made comparisons with SOTA difficult. While authors added HumanML3D in the appendix during rebuttal, the initial omission was a significant oversight (PvFL, Dxar, j864).
4. Reviewers noted that the "best" method (JP) still exhibits floating/jittering artifacts and requires inverse kinematics (IK) for downstream use (e.g., SMPL), which introduces errors and computational cost, a trade-off not fully justified by the authors (Dxar, j864, Dy62).

In summary, this paper was reviewed by four experts in the field. The recommendations are 2, 2, 4, 6. The reviewers raised some serious concerns as pointed out above. After rebuttal, while the paper has merit, the core concern that the paper revisits established findings without comparing against or acknowledging the shift toward latent-based generation methods was not fully resolved. Most reviewers believe that the contribution does not meet the bar for ICLR.

**Reviewer Concerns:**

**Well addressed:**

None

**Partly addressed:**

- Dataset: Reviewers (PvFL, j864, Dxar) criticized the exclusive use of HumanAct12 and 100STYLE. The authors added results on HumanML3D in the Appendix and supplementary video during the rebuttal. However, as this is the standard benchmark, reviewers likely expected it to be central to the main paper's analysis rather than an addendum.


**Unsolved:**

- Limited Novelty / Contribution (PvFL, Dxar): Reviewers pointed out that the findings (JP > Rotations, v-pred > x0) duplicate conclusions from existing works (MARDM, InterGen, ACMDM). The authors' differentiation was not convincing enough to overcome the perception of the work being incremental.
- Lack of Comparison with Latent Diffusion Models (Dxar, j864): Reviewers argued that the field has moved towards Latent Diffusion (MLD, MotionLCM), and findings on raw MDM might not transfer. The authors argued that studying raw motion is still valuable, but reviewers specifically requested comparisons or validation that these insights hold in latent spaces, which was not provided.
- Visual Quality Issues (Dxar): Reviewer Dxar noted that even the best configuration (JP + v-pred) exhibited floating and jittering in the video, which remains a quality concern.

**Reviewer Scores:**

**PvFL (2)**

The reviewer would likely maintain the score. While the authors added HML3D results, the primary critique regarding the "incremental" nature of the findings compared to existing literature remains unaddressed.

**Dxar (2)**

The reviewer would likely maintain the score (2). The reviewer explicitly stated that without results on standard benchmarks and comparisons to latent space models, the validity is difficult to assess. The rebuttal did not fundamentally change the scope to include Latent Diffusion comparisons.

**j864 (4)**

The reviewer would likely maintain the score. The reviewer emphasized the "limited methodological breadth" (only MDM) and the lack of combined representations.

**Dy62 (6)**

The reviewer would likely maintain the score. This reviewer was already positive. But given the strong consensus against the paper from others, Dy62 might lower confidence slightly, but the score would likely remain positive.

---

### Decision · Program_Chairs · 2026-01-26

Reject